

# Brief Communication: Glacier mapping and change estimation using very high resolution declassified Hexagon KH-9 panoramic stereo imagery (1971-1984)

Sajid Ghuffar[1,2], Owen King[1], Grégoire Guillet[1], Ewelina Rupnik[3], and Tobias Bolch[1]

[1]School of Geography and Sustainable Development, University of St Andrews, St Andrews, UK
[2]Department of Space Science, Institute of Space Technology, Islamabad, Pakistan
[3]LASTIG, Univ Gustave Eiffel, ENSG, IGN, F-94160 Saint-Mande, France

**Correspondence:** Sajid Ghuffar (sghuffar@gmail.com), Tobias Bolch (tobias.bolch@st-andrews.ac.uk)

**Abstract.** The panoramic cameras (PC) onboard Hexagon KH-9 satellite missions from 1971-1984 captured very-high resolution stereo imagery with up to $60\,\mathrm{cm}$ spatial resolution. This study explores the potential of this imagery for glacier mapping and change estimation. We assess KH-9PC imagery using data from KH-9 Mapping Camera (MC), KH-4PC as well as SPOT and Pléiades satellites. The high resolution of KH-9PC leads to higher quality DEMs, which better resolve the accumulation region of the glaciers in comparison to the KH-9MC. On stable terrain, KH-9PC DEMs achieve an elevation accuracy $<4\,\mathrm{m}$ with respect to SPOT and Pléiades DEMs. While the estimated geodetic mass balance using PC and MC data are similar, the elevation change data show superior spatial coverage and considerably less noise.

## 1 Introduction

Recent studies confirmed that glaciers have been losing mass globally at an accelerated rate over the last two decades and available data indicate global glacier recession at least since the 1960s (Zemp et al., 2019; Hugonnet et al., 2021). Providing reliable estimates of these long-term changes at regional scales still remains a challenge due to data scarcity. For some regions aerial photos provide the basis for long-term information (Mannerfelt et al., 2022; Korsgaard et al., 2016) but for most regions no historial images exist or are not available. Declassified data from the US satellite reconnaissance program, especially the KH-9 Mapping Camera (MC) and KH-4 Panoramic Camera (PC), have emerged as a key data source for mapping the state of the glaciers from 1960s-1980s (Bhattacharya et al., 2021; Dehecq et al., 2020; Zhou et al., 2017; Pieczonka and Bolch, 2015). However, poor contrast and texture in this imagery leads to large data gaps in the corresponding elevation datasets, which hinders an accurate investigation of glacier changes. Recent availability of declassified very-high resolution (VHR) Hexagon KH-9 panoramic stereo imagery offers further opportunities for improved glacier mapping for the 1970s and 1980s due to its very-high spatial resolution and distinct ground coverage.

The US satellite reconnaissance program designed for strategic surveillance during the Cold War had its first successful launch and film recovery in 1960 involving a Corona KH-1 series mission. Parallel to the development of the Corona program with cameras having the highest spatial resolution of $1.8\,\mathrm{m}$, Gambit-1 KH-7 (1963-1967) and Gambit-3 KH-8 (1966-1984)



programs involved the development of a VHR (60-90 cm) camera system to acquire detailed information about specific targets. The Hexagon program, consisting of 20 missions from 1971-1986, aimed to provide Corona-type coverage together with high resolution of the Gambit program (NRO, 2011). The main camera system in Hexagon KH-9 missions consisted of two PCs with a $20°$ stereo convergence angle and the highest spatial resolution of around $60 \, \mathrm{cm}$ (NRO, 1968), while the later missions (1205 - 1216) also included a frame MC with a nadir looking configuration capable of stereo and tri-stereo overlaps and spatial resolution of 6-9m (Burnett, 2012).

The Corona KH-4PC imagery (declassified in 1995) and Hexagon KH-9MC imagery (declassified in 2002) have played a pivotal role in the estimation of long-term glacier changes (Bhattacharya et al., 2021; Maurer et al., 2019). The processing of declassified photographic films requires special considerations due to the presence of film distortions and limited information of camera parameters. The modeling of panoramic imaging geometry further complicates the processing of PC imagery. While the majority of the earlier work on KH-4PC and KH-9MC has been limited in terms of number of images used or area covered, automated pipelines for both KH-9MC and KH-4PC have recently been proposed, which enables large-scale mapping (Ghuffar et al., 2022; Dehecq et al., 2020; Maurer and Rupper, 2015).

The VHR KH-9PC data was declassified more recently (in 2011), consisting of more than 670,000 scenes covering the majority of the Earth's land area with multiple acquisitions over most glacierized regions of the world (Fig. S1). While the first 1,700 rolls of the KH-9PC imagery were released through the U.S. Geological Survey's (USGS) EarthExplorer in 2015, the scanning of the whole KH-9PC archive and its availability through the USGS is still underway with completion aimed at 2022. The potential for high resolution DEM generation and mapping using KH-9PC has remained largely unexplored until now (Zhou et al., 2021; Fowler, 2016). The aim of this study is to evaluate the potential of this imagery in context of glacier mapping and change estimation. We aim to quantify the accuracy of glacier DEMs generated from KH-9PC and assess the changes in the glacier mass balance and its associated uncertainty.

## 2 Hexagon KH-9 Panoramic Cameras

The KH-9PC system, developed by Pelkin-Elmer, consisted of stereoscopic cameras with a $20°$ convergence angle. The cameras scanned in opposite directions with maximum scan angle of $±60°$ in the across track direction. In contrast to Corona KH-4PCs, these cameras had the capability to acquire images in variable scan width and scan center modes. KH-9PC operated with 30, 60, 90 and 120 degree variable scan angles with scan centers of 0, $±15$, $±30$ and $±45$ (Fig. 1). The KH-9PC system had a focal length of $152.4 \, \mathrm{cm}$ (60 inches) with a folded wright optical system. The orbital altitude of the KH-9 missions was typically in the range of 160-250 km. This resulted in a best ground resolution of around $60 \, \mathrm{cm}$ towards the nadir direction. The ground resolution varied within the scan and reached around $3 \, \mathrm{m}$ towards the $60°$ scan angle. The ground coverage for $120°$ scan was approx. $12\,000 \, \mathrm{km}^2$. However, the distortions at high scan angles were considerable, in addition to the lower ground resolution. Consequently, the later KH-9 missions were restricted to a maximum $45°$ scan angle (NRO, 2011).





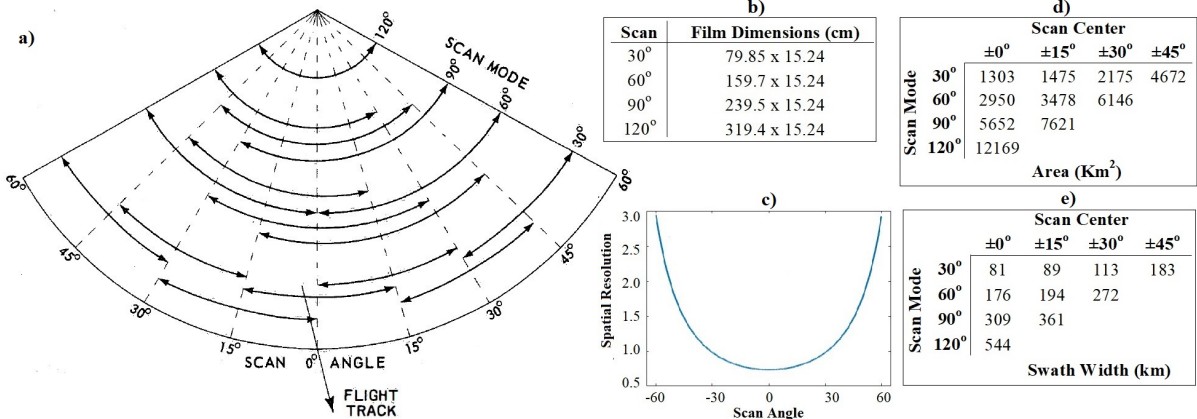

**Figure 1.** a): The KH-9PC acquired images in four different scan angles i.e. 30°, 60°, 90°, 90° along with with different scan centers i.e. ±0°, ±15°, ±30°. b): Film dimensions for each scan angle range. c): Spatial resolution vs scan angle with scan center at 0°. d,e): The area and swath width of the ground footprint for each combination of scan angle and scan center (Modified from NRO (1968)).

## 3   Data

To evaluate KH-9PC data for glacier mapping and change assessment with respect to contemporary high-resolution satellite imagery, we chose two study sites: Ak-Shirak area in central Tien Shan and Passu Glacier in central Karakoram (Fig. S2 and S3 in supplement). The selection of these study sites was based on the availability of KH-9MC DEMs and orthoimages from earlier studies (Pieczonka and Bolch, 2015; Bolch et al., 2017; Goerlich et al., 2017) as well as the availability of contemporary high resolution satellite stereo imagery over these areas. We compare KH-9PC DEMs with the KH-9MC DEMs from the same

satellite overpass using the high resolution Pléiades and SPOT-6 DEMs (Table S1). We further include KH-4PC data over Ak-Shirak for comparison. The KH-9PC data consist of two *aft* and one *fore* image for the Ak-Shirak area and vice versa for the Passu area. The KH-9PC images of Ak-Shirak had scan angle range of -45° to 45° , while the Passu KH-9PC images had scan angle range of -60° to 30° (Table S1).

To assess the potential of KH-9PC data for large-scale or regional scale mapping of glacier changes, we utilize the entire

swath width of the successive KH-9PC stereo pairs (*fore*: F049-F052 and *aft*: A050-A052), which contains the image subset used for Passu Glacier area. We use the ALOS World DEM AW3D30 (30 m) DEM (Tadono et al., 2015) for the evaluation of the KH-9PC DEM of the successive stereo pairs, and derive estimates of surface elevation change (dH) from the two DEMs to examine the potential for estimating glacier mass balance at a regional extent.



## 4 Methods

### 4.1 Processing of KH-9PC Imagery

The photographic film of each KH-9PC scene is scanned in to several parts by the USGS due to its large size. The film length of KH-9PC scenes depends on the total scan angle, while the width of the film is fixed at $16.7\,\mathrm{cm}$. For a $120°$ scan the film length is $319.4\,\mathrm{cm}$ and is scanned into 14 parts with $7\,\mathrm{\mu m}$ resolution (approx. $456,000 \times 22,000$ pixels). We stitch individual scans using tie points extracted in the overlapping region of the successive scan parts to generate an image of the entire scan,

which is then further processed for DEM and orthoimage generation.

To process the KH-9PC imagery, we follow a workflow similar to the Corona Stereo Pipeline (CoSP) presented in Ghuffar et al. (2022). CoSP uses a modified form of collinearity equations to model the imaging geometry of the panoramic cameras with a scanning mechanism. This panoramic camera model includes additional parameters to model the motion of the camera during the panoramic image scan as well as the image motion compensation mechanism. Following the workflow in CoSP,

we match feature points between KH-9PC imagery and Landsat-7 ETM+ panchromatic images using the deep learning model SuperGlue (Sarlin et al., 2020). These feature points along with the corresponding elevation values derived from AW3D30 constitute the Ground Control Points (GCPs), which are used in the estimation of the camera parameters. The initial approximation of the camera parameters is done using the image corner locations given in the metadata of KH-9PC imagery available from the USGS. These approximate camera parameters are then optimized in a bundle adjustment using GCPs and tie points

of the stereo pair.

To map the corresponding image points of the stereo pair to the same image row, we use the generic epipolar resampling algorithm presented in Deseilligny and Rupnik (2020). Then, we use the semi-global matching algorithm (Hirschmuller, 2007) for dense matching of the resampled stereo image pair. These dense stereo correspondences are then triangulated to generate a 3D point cloud using the estimated camera parameters. To compensate the misalignment between the reference DEM and

the KH-9PC point cloud a tile-based coregistration with the reference DEM is performed using least squares surface matching algorithm employing a 3D affine transformation (Pfeifer et al., 2014). The coregistered KH-9PC point cloud is then interpolated to a raster DEM at the resolution of the reference DEM i.e. $5\,\mathrm{m}$ for Passu, $10\,\mathrm{m}$ for Ak-Shirak and $30\,\mathrm{m}$ for comparison with AW3D30.

### 4.2 KH-9MC, KH-4PC, SPOT-6 and Pléiades DEMs

A Pléiades DEM ($5\,\mathrm{m}$, 2021) of the Passu glacier was generated using MicMac (Pierrot-Deseilligny et al., 2014). We used DEMs of KH-9MC ($25\,\mathrm{m}$ for Ak-Shirak and $30\,\mathrm{m}$ for Passu Galcier), KH-4PC ($25\,\mathrm{m}$, 1964) and SPOT-6 ($10\,\mathrm{m}$, 2017) generated in earlier studies. The KH-9MC images of Ak-Shirak and Passu Glacier were processed in Leica Photogrammetry Suite (Pieczonka and Bolch, 2015) and ERDAS IMAGINE (Bolch et al., 2017) respectively. The KH-4PC images were processed in Remote Sensing Software Package Graz (Goerlich et al., 2017), while the SPOT-6 DEM of Ak-Shirak was generated using

PCI Geomatica (Bhattacharya et al., 2021).



## 4.3 DEM differencing and elevation change post-processing

Coregistered DEMs were differenced from their reference DEMs (i.e. SPOT-6 DEM for Ak-Shirak and Pléiades DEM for Passu). To enable the robust estimation of geodetic glacier mass balance over the Passu and Petrov Glaciers, we firstly remove erroneous elevation change (dH) estimates most common in glacier accumulation zones (Fig. 2). We discard dH values outside $\pm\,200\,\mathrm{m}$ under the assumption that glacier thinning or thickening outside of this range is unlikely. We then further filter the remaining dH data using a threshold of $<\pm\,3*$ the standard deviation of dH within $50\,\mathrm{m}$ elevation bands of glacier surfaces, through the full elevation range of each glacier. We fill resulting gaps in the dH grids using the mean value of dH from the same $50\,\mathrm{m}$ elevation band (McNabb et al., 2019). We convert glacier-wide dH to volume change estimates considering the pixel size of the dH grids, and then to mass change using a conversion factor of $850\,\mathrm{kg\,m^{-3}}$ (Huss, 2013). The uncertainty associated with geodetic mass balance estimates over Petrov and Passu glaciers was calculated following the approach of Fischer et al. (2015) (after Rolstad et al. (2009)), which considers the variance of dH data over stable off-glacier areas as representative of the uncertainty of dH estimates on-glacier, and the uncertainty of the density conversion factor proposed by Huss (2013), when converting ice volume to mass changes. We also follow the approach of Malz et al. (2018) to consider the impact of a changing glacier area on overall glacier mass balance estimates.

## 5 Results

The differencing of KH-9PC DEMs with the SPOT-6 and Pléiades DEMs show an NMAD and 68% (confidence interval) of less than $4\,\mathrm{m}$ over stable terrain (Table 1), while the KH-9MC DEMs show an NMAD of $12.66\,\mathrm{m}$ for Ak-Shirak and $27.28\,\mathrm{m}$ for Passu area. Although the KH-9PC DEMs show significantly better accuracy, it should be emphasized that the quality of the KH-9MC and KH-4PC DEMs (Table 1) are also dependent on the DEM generation workflows adopted in the respective studies. The accuracy of the KH-9MC DEMs reported in earlier studies show significant variation due to processing differences (Dehecq et al., 2020; Bolch et al., 2017; Zhou et al., 2017). The best accuracy reported for KH-9MC data is around $5\,\mathrm{m}$ (68%) with respect to SRTM DEM using the KH-9MC imagery over the European Alps (Dehecq et al., 2020).

Over Petrov Glacier (Ak-Shirak), the mean thinning estimates were similar for dH data derived using the KH-9PC ($-26.8\,\mathrm{m}$) and KH-9MC ($-26.0\,\mathrm{m}$) and the data were similarly dispersed (StDev $28.8\,\mathrm{m}$ for PC, $30.0\,\mathrm{m}$ for MC) (Fig. 2). Over Passu Glacier, the KH-9PC ($-6.2\,\mathrm{m}$) and KH-9MC ($-9.0\,\mathrm{m}$) again produced similar dH estimates between 1973-2021, whilst the StDev of KH-9PC dH data ($17.6\,\mathrm{m}$) were slightly lower than those of the KH-9MC dH data ($23.0\,\mathrm{m}$). Unfiltered dH data (Fig. 2) clearly show the extent of glacier accumulation zones affected by low surface contrast in DEM generation, and resulting blunders caused anomalous, high-magnitude ($-50\,\mathrm{m}$) dH estimates (Fig. 2) over both study sites. The area affected by such errors was much smaller in the KH-9PC DEM than the KH-9MC DEM. Over Passu Glacier 80.2% of KH-9PC dH data was retained after filtering, compared to 56.6% in the case of the KH-9MC dH grid. Over Petrov Glacier, KH-9PC dH data covered 80.2% of the glacier area following filtering, whereas 68.2% of KH-9MC dH data were retained.

The mass balance of Petrov Glacier was estimated to be $-0.41\pm0.04$ m w.e.a[-1] for 1973-2017 when using KH-9PC data. While, using KH-9MC data, we estimate the mass balance of the glacier to be $-0.46\pm0.07$ m w.e.a[-1]. The mass balance of

**Figure 2.** a): dH of KH-9PC (1973), KH-9MC (1973) and KH-4PC (1964) DEMs with the SPOT-6 (2017) DEM over Ak-Shirak. b): dH of the KH-9PC (1973) and KH-9MC (1973) DEMs with Pléiades (2021) DEM over Passu Glacier. Bottom Row: The corresponding dH histograms for glacier and off-glacier pixels and change of dH with respect to the elevation. The linear feature visible in the dh image over the accumulation zone of Passu Glacier is due to the boundary of successive images having a small overlap.



**Table 1.** Mass balance (MB) of Petrov and Passu glaciers using KH-9PC and MC data and dH statistics of stable terrain between the KH-9 and the contemporary DEMs. The dH statistics over stable terrain from Dehecq et al. (2020)* and Zhou et al. (2017)** are given for a comparison.

| | | **Stable Terrain Statistics** | | | | |
|---|---|---|---|---|---|---|
| | Camera | Median | NMAD | SD | 68% | 95% |
| **Ak-Shirak** | **KH-9PC** | -0.06 | 3.35 | 8.32 | 3.55 | 15.68 |
| | **KH-9MC** | 0.82 | 12.66 | 18.37 | 13.64 | 39.43 |
| | **KH-4PC** | 0.10 | 9.30 | 17.14 | 9.75 | 37.89 |
| **Passu** | **KH-9PC** | -0.20 | 3.54 | 11.44 | 3.80 | 19.95 |
| | **KH-9MC** | -0.35 | 27.78 | 35.53 | 31.60 | 78.06 |
| * | **KH-9MC** | - | - | - | 5.00 | 15.00 |
| ** | **KH-9MC** | 0.04 | 19.14 | 22.61 | - | - |

| **Petrov Glacier MB** | | **Passu Glacier MB** | |
|---|---|---|---|
| **1973-2017 (m w.e.a/yr)** | | **1973-2021 (m w.e.a/yr)** | |
| **KH-9PC** | **KH-9MC** | **KH-9PC** | **KH-9MC** |
| **-0.41 ± 0.04** | **-0.46 ± 0.07** | **-0.09 ± 0.04** | **-0.1 ± 0.14** |

Passu Glacier was estimated to be $-0.09 \pm 0.04$ m w.e.a$^{-1}$ for 1973-2021 based on KH-9PC data. While, using KH-9MC data,

the estimated mass balance was $-0.10 \pm 0.14$ m w.e.a$^{-1}$.

## 5.1 Towards large-scale mapping with KH-9PC

The differencing of AW3D30 with DEMs derived from successive KH-9PC stereo pairs (using the entire image i.e. $-60°$ to $30°$ scan angle) shows that the systematic errors are relatively low up to $45°$ scan angle (Fig. 3). The NMAD of elevation differences over stable terrain is around $5.5$ m from -45° to 30° scan angle, which shows that successive stereo DEMs are well

coregistered and consistent with each other. However, the overlap between the successive frames is relatively small i.e. around 3% towards the nadir, which may lead to data gaps at the boundary of successive DEMs especially if the boundary falls over a texture-less image area. Most moderate to large-size glaciers require mosaicing of multiple successive KH-9PC DEMs and fine coregistration and filtering of any boundary artifacts are important to avoid introducing bias in the DEMs.

At higher scan angles (i.e $> 45°$) the occlusions (from mountains due to slant viewing angle) increase significantly and

higher perspective distortions lead to systematic bias in the DEM differences due to uncompensated systematic errors (Fig. 3). Even with an improved coregistration and bias/trend correction approach, reliable DEM and orthoimage generation for higher scan angles will be quite challenging especially for mountainous regions. Therefore, the scan angle range of the KH-9PC acquisitions should always be considered, when investigating the ground coverage of KH-9PC images.



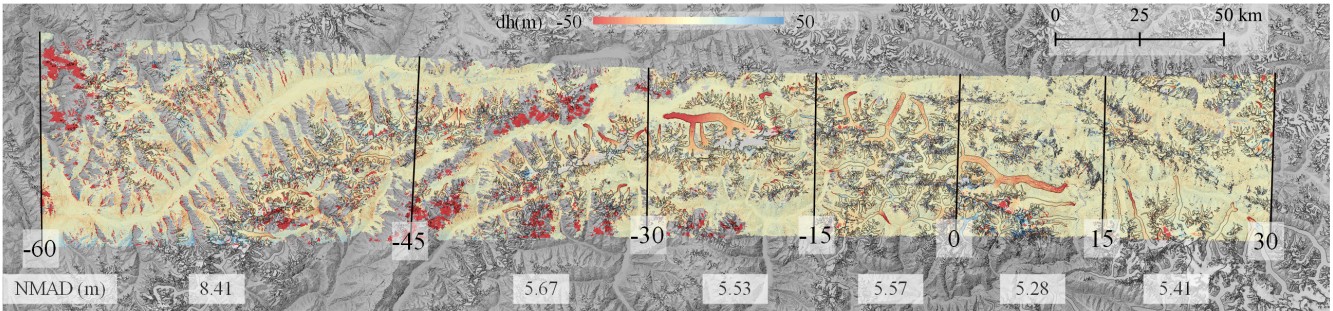

**Figure 3.** Elevation differences using the entire KH-9 panoramic swath width of successive KH-9PC stereo images D3C1206-200215F049-F52, D3C1206-200215A050-A052 and the AW3D30 covering Karakoram and Hindukush mountains. The near vertical lines show $15°$ intervals of the panoramic scan. The NMAD of the elevation differences over stable terrain are shown at the bottom for each $15°$ scan area. Occlusions due to high relief and slant viewing angle cause data gaps between -60° to -35° scan angle, while some data gaps and outliers are due to clouds (see Fig. S4).

## 6 Discussion

The accuracy of optical stereo derived elevation data depends on the image resolution and the results show that KH-9PC data can achieve better accuracy in comparison to KH-9MC and KH-4PC, reducing uncertainties associated with the estimation of glacier surface elevation change. Our results demonstrate the ability of the KH-9PC to better resolve surface conditions in glacier accumulation zones, therefore providing greater coverage of dH data to be used in the study of glacier accumulation and ablation processes (Figs. 2 and S5). Geodetic studies incorporating KH-9MC data (e.g. (Zhou et al., 2017; King et al., 2019;

Maurer et al., 2019)) report considerable data gaps (up to 40%), primarily over the higher reaches of glaciers, and therefore provide little information on their accumulation regime over multi-decadal time periods. In combination with contemporary sensors capable of capturing accumulation zone conditions (e.g. Pléiades), KH-9PC data can be used to more completely capture glacier accumulation and ablation processes over longer time periods than is currently possible. In our two case studies, this improved coverage resulted in relatively minor differences in geodetic mass budgets (Table 1), likely because of the

effectiveness of outlier filtering and subsequent gap filling techniques on minimizing the impact of erroneous dH estimates on glacier volume change (and therefore mass balance) calculations (e.g. (McNabb et al., 2019)). Still, the improved coverage afforded by KH-9PC data over the higher reaches of mountain glaciers could be significant when examining processes such as quiescent phase ice mass build-up over surge-type glaciers, or concentrated ice mass accumulation prior to glacier instabilities (Kääb et al., 2018).

The VHR KH-9PC imagery complements the data from KH-9MC and KH-4PC in terms of area coverage and resolution and offers potential for improved glacier mapping capabilities. Although the radiometric resolution of scanned KH-9PC imagery is less as compared to Pleiades and SPOT-6 (8-bits vs 12-bits), the spatial details are quite similar to contemporary VHR satellite imagery (Fig. S6) and therefore has the capability of characterising small scale glacier surface features such as supraglacial ponds and ice cliffs. This improves our ability to study the role of these ablative features over longer timescales than is currently



possible. The different ground footprint of KH-9PC (Fig. S2) as compared to KH-9MC as well as acquisitions from higher number of KH-9 missions also enhances the chances of identifying and mapping glacier surge events from 1970s to 1980s (see Fig. S7, which shows the Hispar Glacier (Karakoram) tributary surge in multi-temporal KH-9PC imagery). The crevasse pattern indicative of a surge activity are also recognizable in VHR KH-9PC imagery. In addition, the KH-9PC imagery enhances the potential for mapping of glacial lake outburst flood events (from 1970s to 1980s) in high spatial detail (Fig. S7).

The presence of film distortions due to long-term storage poses a limitation in processing historical imagery. The reseau marks in the KH-9MC film enables correction of the film distortion. However, such reseau grid is not available in PC due to distinctive imaging mechanism. To the best of our knowledge, there exists no established methodology for correction of film distortions in declassified panoramic imagery. In addition, the scanning artifacts have also been reported in the scanned imagery, which may further limit the accuracy of the derived data (Ghuffar et al., 2022; Dehecq et al., 2020).

## 180  7  Conclusions

This study shows that KH-9PC DEMs can achieve accuracy better than 4m over mountainous terrain, which is an improvement on the accuracy reported for KH-9MC and KH-4PC DEMs. The very high resolution and better image quality leads to more reliable elevation estimates towards the accumulation region of the glaciers. The estimated geodetic mass balance using PC data ($-0.09 \pm 0.04$ m w.e.a$^{-1}$ for Passu Glacier and $-0.41 \pm 0.04$ m w.e.a$^{-1}$ for Petrov Glacier) and MC data ($-0.10 \pm 0.14$

m w.e.a$^{-1}$ for Passu Glacier and $-0.46 \pm 0.07$ m w.e.a$^{-1}$ for Petrov Glacier) are quite similar, which shows the effectiveness of the outlier filtering approach. However, the uncertainty associated with mass balance estimate is significantly reduced when considering KH-9PC data due to fewer outliers in derived dH data and lower height differences over stable terrain. As cloud free acquisitions in declassified imagery during late summer to early winter months over glacierised areas are rather limited, the VHR KH-9PC imagery with variable footprint offers multiple benefits in the context of glacier mapping and change estimation.

*Data availability.* The utilized data and resulting DEM differencing grids are available from the first author upon request. Certain licence restrictions may apply to satellite imagery. The KH-9PC and MC, KH-4 and Landsat-7 ETM+ images can be ordered and downloaded from https://earthexplorer.usgs.gov/ and the ALOS data from https://www.eorc.jaxa.jp/ALOS/en/index_e.htm.

*Author contributions.* SG and TB designed the study. SG supported by ER processed the KH-9 Panoramic Camera images. OK and GG computed the mass balance and uncertainty estimates. SG supported by TB, OK and GG wrote the manuscript. All authors contributed to
the final form of the article.

*Competing interests.* TB is a member of the editorial board of The Cryosphere. The authors have no other competing interests to declare.





*Acknowledgements.* This study was supported by the Strategic Priority Research Program of Chinese Academy of Sciences (XDA20100300) and the Swiss National Science Foundation (200021E_177652/1).



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
