# Peer review of "Brief Communication: Glacier mapping and change estimation using very high resolution declassified Hexagon KH-9 panoramic stereo imagery (1971-1984)"

_The Cryosphere, 2022_

## Author Response (AR1)

We would like to thank the editor for a positive feedback on our review response letter and have improved the text based on the reviewer comments. These changes have been highlighted and documented below with respect to each reviewer comment. The changes have also been highlighted in the revised manuscript.

**Comments Reviewer 1: Amauary Dehecq**

The study by Ghuffar et al. aims at applying a previously published methodology to a different satellite data set. The methodology was developed to generate Digital Elevation Models from film-based declassified satellite images. While the previously published paper was focused on images from the Corona (KH-4) satellite series, this study focuses on the Hexagon panoramic camera images (KH-9 PC). There was apparently no novel methodological development required for this data set, but because the dataset studied here (KH-9PC) has been barely exploited so far, this study is a valuable contribution.

I only have minor comments, mostly intended to clarify the text or provide additional results, before the article can be published.

We thank the reviewer for the positive evaluation. We agree with the reviewer comments and highlight that the main aim of this paper is to show the immense potential of the KH-9 very high resolution panoramic imagery.

**Minor comments:**

**- Preprocessing (section 4.1): I understand that the photogrammetric part follows exactly the method of Ghuffar et al. (2022). However, in that study, there is also an extensive preprocessing part in order to correct for film distortion using edge markers (rail holes, PG stripes…). Here you only mention the stitching of the image part. As for the KH-4 and KH-9 MC images, I would expect that some distortion exist, due to film distortion and due to the scanning. Can you please elaborate a bit more on the preprocessing? If there is no other processing step required, I would explain why as it seems rather surprising. If you**

**applied a preprocessing step, please detail it. Are there any markers on the image that can help identify and correct for film distortion?**

The KH-9 PC Imagery also contains markers similar to the KH-4B imagery on the both edges along the film length. These markers include scan angle marks, stripes, timing marks as well as titling information such as image Id and acquisition date. Similar to the Corona KH-4 missions these additional markers show some variation from earlier to later missions e.g. the stripes on the both edges is not available on all images. A thorough evaluation of the film distortion in the KH-9 PC is beyond the scope of this study and is expected to be part of a future contribution. So, in this work we do not correct film distortions.

We have added the following text in the Section 4.1 Processing of KH-9PC Imagery:

**"The scanned film consists of the imaged area as well as reference data such as the timing marks, scan angle marks, and titling information. We clip the imaged area and align the length and width of the film along the horizontal and vertical axes of the image, while no film bending estimation and subsequent compensation nor radiometric correction are applied to the image."**

**- DEM coregistration: at L 102, you mention the use of "coregistered DEMs". You describe the tile-based coregistration used for the KH-9PC DEMs, but it is not clear what coregistration method you used for the other DEMs. Do you apply the same tile-based coregistration or did you use the coregistration method of the previous studies? If so, can you explain here what the method was?**

Only the KH-9 PC DEMs have been coregistered using the tile based technique. The earlier DEMs have been coregistered with the method suggested by Nuth and Kääb [1].

We have included the following text in Section 4.2: KH-9MC, KH-4PC, SPOT-6 and Pleiades DEMs

**The fine coregistration of these DEMs have been performed using method of Nuth and Kääb (2011) and the biases due to tilt in the DEMs have been removed using polynomial trend surfaces.**

**[1] Nuth, C. & Kääb, A. Co-registration and bias corrections of satellite elevation data sets for quantifying glacier thickness change. *Cryosphere* 5, 271–290 (2011).**

**- Results: In Figure 2, for Passu glacier, there seem to be a 50% difference (-6.2 vs -9.0 m) between the mean elevation change calculated from both datasets (KH-9PC and KH-9MC), but the mass balance values report in Table 1 and later in the text differ by less than 10%. Can you please explain? I believe this might be because the values are on the unfiltered results? In that case, I would suggest in that Figure 2 to also show the results after filtering and gap-filling.**

The reviewer is correct in their interpretation that the differences in the mean elevation change values originate from the filtering/gap filling process. The difference is found between the filtered (but not yet gap-filled) dH data. The KH-9MC dH grid has many more gaps over the accumulation area of Passu Glacier following filtering, so the mean value of dH is biased by the substantial thinning seen over the lowermost ~3 km of the glacier. The coverage of the KH-9PC dH grid is much more complete over the upper reaches of the glacier, so the mean dH value is less negative. Following gap-filling, the mean values of dH of the KH-9MC Vs Pleiades and KH-9PC Vs Pleiades dH grids are much closer (-6.06 KH-9MC, -5.44 KH-9PC Vs Pleiades) and so the subsequently derived mass balance estimate are also closer.

We have added a separate figure i.e. Figure S5 in the supplement, which shows the dH grids after outlier filtering and gap filling. We have also modified the caption of Fig. 2 to clarify that the elevation changes shown here are for the raw dH grids i.e. without the outlier filtering and gap filling. Modified caption of Fig.2 is:

**"Figure 2. a): dH of KH-9PC (1973), KH-9MC (1973) and KH-4PC (1964) DEMs with the SPOT-6 (2017) DEM over Ak-Shirak. b): dH of the KH-9PC (1973) and KH-9MC (1973) DEMs with Pleiades (2021) DEM over Passu Glacier. Bottom Row: The corresponding dH**

**histograms for glacier and off-glacier pixels and change of dH with respect to the elevation. These dH grids and the corresponding histograms show elevation differences without the filtering and gap filling, which is performed before mass balance computation. The filtered version of the dH grids is available in the Fig. S5 in the supplement. The linear feature visible in the dH image over the accumulation zone of Passu Glacier is due to the boundary of successive images having a small overlap."**

**Specific comments:**

**- L12: Mannerfelt et al. (2022) is not about aerial images. Either rephrase the sentence or use a different citation (e.g., Girod et al. (2018), Geyman et al. (2022))**

We have changed the citation to Geyman et al. (2022)) as suggested

**- L24: "with high resolution" → "with the high resolution"**

This has been corrected.

**- Figure 1, caption – two typos: "90°, 90°" should be "90°, 120°" and "with with".**

This has been corrected.

**- section 4.2: I am amazed by how many different software you are able to leverage!**

**Thank you.**

**- L109: "We convert glacier-wide dH to volume change estimates considering the pixel size of the dH grids". Note that for volume calculation, it is more accurate to calculate a mean dH of all pixels, then multiply by the glacier polygon area, rather than using the pixel count and area, which is more discretized. Of course, it does not matter too much in this study since the focus is not on the glaciological interpretation of the results, and since the same method is applied in all cases, they are directly comparable.**

We thank the reviewer for highlighting this alternative approach. To check the consistency of our original method, we calculated our mass balance estimates using this mean dH approach described by the reviewer and find only minor differences across the various datasets which are well within the estimated uncertainty. Mass balance estimates are within 0.01 m w.e.a$^{-1}$ at both study sites, apart from the KH-9 mapping camera Vs Pleiades derived mass balance estimate for Passu Glacier, which is 0.04 m w.e.a$^{-1}$ more negative using the mean dH approach (-0.10 Vs -0.14 m w.e.a$^{-1}$). As these minor differences do not impact our interpretation, we have retained our original results.

**- L110: Can you state which error correlation length was used in the Fischer et al formula? Note that this formula tends to largely underestimate uncertainty, and I can only recommend to follow the approach of Hugonnet et al. (2022).**

We have added the correlation lengths in section 4.3: DEM differencing and elevation change post-processing.

**"The calculated correlation length was 605 m in the case of the KH-9PC dH data, and 873 m in the case of the KH-9MC data over the Passu Glacier. The correlation length was 1488 m in the case of the KH-9PC derived dH data and 1220 m for the KH-9MC derived dH data over the Petrov Glacier."**

We are aware of the alternative approach of Hugonnet et al. (2022) but feel that this approach, or indeed any suitable method of estimating the uncertainty associated with the elevation change data, will produce essentially the same observations that we already have. More specifically, our aim here is to show that the uncertainty associated with the KH-9MC derived elevation change data is higher than that derived from KH-9PC data. The magnitude of this difference may slightly differ with alternative approaches, but we believe we have already achieved our aim of showing how they contrast.

**- Figure 3: Would you be able to show the approximate footprint of each image on the figure? This would help interpret the small steps in the DEM visible on the left.**

We have added the footprint of the scene D3C1206-200215A051to the Figure 3. We have not added footprints of all the scenes as this will add many polygons to the image, which will make it harder to interpret.

References:

Geyman, E.C., J. J. van Pelt, W., Maloof, A.C., Aas, H.F., Kohler, J., 2022. Historical glacier change on Svalbard predicts doubling of mass loss by 2100. Nature 601, 374–379. https://doi.org/10.1038/s41586-021-04314-4

Girod, L., Nielsen, N.I., Couderette, F., Nuth, C., Kääb, A., 2018. Precise DEM extraction from Svalbard using 1936 high oblique imagery. Geoscientific Instrumentation, Methods and Data Systems 7, 277–288. https://doi.org/10.5194/gi-7-277-2018

Hugonnet, R., Brun, F., Berthier, E., Dehecq, A., Mannerfelt, E.S., Eckert, N., Farinotti, D., 2022. Uncertainty analysis of digital elevation models by spatial inference from stable terrain. IEEE Journal of Selected Topics in Applied Earth Observations and Remote Sensing 1–17. https://doi.org/10.1109/JSTARS.2022.3188922
Citation: https://doi.org/10.5194/tc-2022-203-RC1

**Anonymous Referee #2**

In this manuscript, the authors have presented a comparison of results of geodetic mass changes obtained by applying existing methods to a largely untapped resource. The results are well-presented and very promising, and the potential of this method for extending and improving observations of geodetic mass changes into the past is very exciting. I only have a few small comments/questions that should be considered before the paper is accepted for publication.

Thank you very much for the overall positive evaluation

Section 4.1: After stitching the images, do you use the image frame or rail holes to resample/align the fore and aft images? Do you crop the image border? A little bit more information about the process here would be helpful.

Yes, we crop the imaged area from the film and use it for further processing. However, we do not perform any bending correction using the markers in this work. We have added the following text in the Section 4.1 Processing of KH-9PC Imagery to clarify this point:

**"The scanned film consists of the imaged area as well as reference data such as the timing marks, scan angle marks, and titling information. We clip the imaged area and align the length and width of the film along the horizontal and vertical axes of the image, while no film bending estimation and subsequent compensation nor radiometric correction are applied to the image."**

**Are the different scanned parts of the images radiometrically similar, or do you need to balance/blend the parts together?**

We do not perform any radiometric corrections to the individual scan parts. Visually they appear quite similar. The feature extraction and dense matching is robust to global illumination changes, therefore, we do not expect significant differences due to any slight radiometric differences.

We have mentioned this in the text as given in the above comment.

**In addition to the references, could you mention how/where each of the different steps described here are implemented? (e.g., software/programming languages)?**

We have used the same set of softwares/programming languages as was used in Corona Stereo Pipeline [1] i.e. a combination of Python (Feature Matching), MATLAB (bundle adjustment), MicMac (Epipolar Resampling and dense matching) and OPALS (DEM coregistration).

We have modified the text in the Section 4: Processing of KH-9PC Imagery to clarify this:

**"To process the KH-9PC imagery, we follow a workflow similar to the Corona Stereo Pipeline (CoSP) using the same set of software and libraries as presented in Ghuffar et al. (2022)."**

[1] Ghuffar, S., Bolch, T., Rupnik, E., & Bhattacharya, A. (2022). A Pipeline for Automated Processing of Declassified Corona KH-4 (1962–1972) Stereo Imagery. *IEEE Transactions on Geoscience and Remote Sensing*, *60*, 1-14.

**Section 4.3: l. 110: what correlation length did you use for the Fischer et al. formula?**

We have added the correlation lengths in section 4.3: DEM differencing and elevation change post-processing.

**"The calculated correlation length was 605 m in the case of the KH-9PC dH data, and 873 m in the case of the KH-9MC data over the Passu Glacier. The correlation length was 1488 m in the case of the KH-9PC derived dH data and 1220 m for the KH-9MC derived dH data over the Petrov Glacier."**

---

## Author Response (AR2)

We thank the editor and the reviewers for the positive feedback as well as thoughtful suggestions/corrections. We have updated the manuscript accordingly.

Dear Sajid and co-authors,

I have reviewed your revised manuscript and am pleased that it is now accepted for publication in our journal, pending some technical corrections (see below). Congratulations, and many thanks for your valuable contribution to The Cryosphere!

Kind regards

Bert Wouters

- Line 39-40, "the scanning of the whole KH-9PC archive and its availability through the USGS is still underway with completion aimed at 2022". Meanwhile, we entered 2023, please update the status/planning of the scanning.

We have now added the recent status that we received from the USGS. i.e. By the end of February 2023, USGS has scanned about 70\% of the browse images with the completion aimed at early 2024 (USGS, pers. comm.).

- Line 40: "wright" -> "Wright"

corrected

- Line 113-118: "... the variance of dH data over stable off-glacier areas as representative of the uncertainty of dH estimates on-glacier, and the uncertainty of the density conversion factor proposed by Huss (2013), when converting ice volume to mass changes. The calculated correlation length was 605 m in the case of the KH-9PC dH data, and 873 m in the case of the KH-9MC data over the Passu Glacier. The correlation length was 1488 m in the case of the KH-9PC derived dH data and 1220 m for the KH-9MC derived dH data over the Petrov Glacier.": This is just a suggestion, but it would make sense to mention the correlation lengths before you mention the density conversion factor, since the two are not related. I would also recommend to rephrase the last sentence to avoid repetition.

E.g.: ... the variance of dH data over stable off-glacier areas as representative of the uncertainty of dH estimates on-glacier. The calculated correlation length was 605 m in the case of the KH-9PC dH data, and 873 m in the case of the KH-9MC data over the Passu Glacier. For Petrov Glacier, correlation lengths of 1488 m and 1220 m were found for KH-9PC KH-9MC derived dH data, respectively. We include the uncertainty in the density conversion factor proposed by Huss (2013), when converting ice volume to mass changes and also follow the approach of Malz et al. (2018) "

Revised as suggested

- Supplement figure S2: "only one pair of KH-4 image footprints ARE shown here" -> "only one pair of KH-4 image footprints IS shown here" (pair is singular)

corrected